

# Stochastic process drives the dissimilarity in biodiversity patterns between *Pinus kwangtungensis* coniferous forest and evergreen deciduous broad-leaved mixed forest in karst area

Xingying Fan[1,*], Longchenxi Meng[2,*], Yeheng Wang[3] and Lipeng Zang[2,4]

[1] School of Medicine and Health Management, Guizhou Medical University, Guiyang, Guizhou Province, China

[2] Research Center of Forest Ecology, College of Forestry, Guizhou University, Guiyang, Guizhou Province, China

[3] Zibo Real Estate Registration Center, Zibo, Shandong Province, China

[4] Guizhou Libo Karst Forest Ecosystem National Observation and Research Station, Libo, Guizhou Province, China

[*] These authors contributed equally to this work.

## ABSTRACT

*Pinus kwangtungensis* is an endangered evergreen conifer tree species, and its *in situ* conservation has been considered one of the most critical issues. However, relative protection is limited by the lack of understanding of its community structure and underlying assembly processes. To study how the species diversity and assembly processes of *Pinus kwangtungensis* coniferous forest (CF) differed with regional climax community, this study established a series forest dynamic plots both in CF and evergreen deciduous broadleaved mixed forest (EDBM). By performing comparison analysis and PER-SIMPER approaches, we quantified the differences in species diversity and community assembly rules. The results showed that the species α-diversity of CF differed greatly from the EDBM both in species richness and evenness. In addition, the stochastic process acted a more important role in determining species composition, indicating the uncertainty in presence of species. The soil phosphorus and changeable calcium content were the main factors driving the differences in biodiversity, which the importance of soil nutrient factors in driving species composition. Our study highlighted that we should consider the community structure and ecological process when conducting conservation of *Pinus kwangtungensis*.

# INTRODUCTION

High species extinction rates emphasize the importance of biodiversity conservation (*Chen et al., 2021*). However, relative protection is limited due to a lack of understanding of biodiversity patterns and underlying maintenance mechanisms, especially in those vegetation types distributed in special habitats (*Alder et al., 2002*; *Feng et al., 2021*). *Pinus*

Corresponding author
Lipeng Zang, cafzanglp@163.com

*kwangtungensis*, an endangered evergreen conifer tree species, is unique to the southern tropical zone of China. Due to its sensitivity to temperature and precipitation, its native habitat is limited to the southern subtropical zone or the upper part of the tropical northern edge (*Wang & Cui, 2023*). *P. kwangtungensis* coniferous forest (CF) typically comprises one dominant species, with high species richness in the understory (*Cui et al., 2012*). How to perform *in situ* conservation on such a unique community has been considered an essential issue in biodiversity conservation.

Understanding the community structure and species composition is the basis of biodiversity conservation since they are determined by confusing factors such as interspecific competition or stochastic processes (*Feng et al., 2021*; *Magurran, 2021*). Furthermore, biodiversity patterns encompass not only the number of species but also species evenness, indicating species distribution among different communities (*He, Legendre & Lafrankie, 2010*; *McGill, 2011*). Previous studies have focused on species richness while neglecting differences in evenness (*Chao, Chiu & Jost, 2014*; *Ulrich et al., 2016*). Hence, it is necessary to combine richness and evenness to better represent species diversity patterns, which emphasizes the importance of distinguishing the species composition (*Matthews, Borges & Whittaker, 2014*; *McGill et al., 2007*). The species abundance distribution (SAD) curve can display both richness and evenness, providing a visual representation of differences in species diversity patterns among communities (*Chisholm et al., 2014*; *Xue et al., 2021*). Since more than one mechanism can shape the SAD curve, quantifying the differences in SAD components can better display the biodiversity pattern and underlying processes (*Muhic et al., 2023*). For instance, a steep slope of the SAD curve usually alludes to species dominance or commonness, while a long tail of SAD indicates the species rarity (*Magurran, 2021*; *McGill et al., 2007*). Species dominance is to what extent the common species predominates the community, while the species rarity reflects the proportion of rare species in community abundance (*Simons et al., 2017*). Combining the species dominance and rarity can clearly describe species alpha diversity (α-diversity) among communities at regional scales.

Actual communities are formed by complicated ecological processes such as habitat filtering, limiting similarity, or stochastic processes (*Chase, 2010*; *Franklin et al., 2013*). Each ecological process influences the species composition and biodiversity patterns. For example, regional species pools can impact community composition, as studies have demonstrated a significant positive correlation between regional species pools and community species richness, serving as the foundation for determining community composition (*Zobel, 1997*). Interspecific interactions determine the survival and relative abundance of each given species in a local microhabitat (*Burton et al., 2011*). Abiotic factors reflect the strength of habitat filtering and is a crucial limiting factor in determining species occurrence, which play an important role in shaping community assembly (*Su et al., 2023*). The niche-based framework suggested that communities were equilibrium assemblages in which interspecific competition for limited resources and other biotic interactions determine the presence of species (*Armas, Rodríguez-Echeverría & Pugnaire, 2011*; *Münkemüller et al., 2020*; *Ulrich et al., 2016*; *Zhou & Wang, 2023*). On the other hand, the neutral-based assembly framework emphasized that community structure resulted

from purely stochastic ecological drift, ultimately leading to ecologically equivalent species resulting from random dispersal and extinction (*Hubbell, 2001*; *Loke & Chisholm, 2023*). Recent studies preferred combining the two conflicting perspectives by linking stabilization mechanisms within species and fitness similarities among species after debating for over 20 years (*Chesson, 2000*; *Leibold et al., 2022*; *Münkemüller et al., 2020*). What differed was that which process acted as the first-order driver in the actual community (*Loke & Chisholm, 2023*; *Münkemüller et al., 2020*). Since SAD could well reflect the underlying mechanism driving the community structure (*McGill et al., 2007*; *Ulrich et al., 2016*), SAD model fitting became a more popular approach to reveal the underlying community assembly rules of local taxonomy, and was used to explain why and to what extent the species composition varied among communities (*Gibert & Escarguel, 2019*; *Magurran, 2021*; *Matthews et al., 2019*; *Zhang et al., 2016*).

Previous studies examined the community assembly rules mainly by performing SAD fitting on each community and comparing the significance of deviation from the null models to identification of the main first-order driver (*i.e.,* niche- or dispersal-assembly processes) (*Chase & Myers, 2011*; *Lim, Fine & Mittelbach, 2015*; *Münkemüller et al., 2020*). However, typical SAD fitting approaches based on α-diversity could not directly examine which process determined the difference in species composition between two communities. *Gibert & Escarguel (2019)* developed the permutation-based algorithm building on Clarke's similarity percentage (PER-SIMPER) (*Clarke, 1993*) to directly examine which first-order assembly dominates the dissimilarity between communities. An empirical SIMPER profile is the relative contribution of each taxon to the overall average dissimilarity (OAD) between two or more groups of taxonomic assemblages, depicting the contribution of each taxon to the mean inter-group beta diversity (β-diversity) compared with intra-group β-diversity (*Clarke, 1993*). The PER-SIMPER, which combined the empirical SIMPER profile with null SIMPER profiles generated from permutation on the data set, could well distinguish the first-order assembly driving the dissimilarity between groups of taxonomic assemblages (*Gibert & Escarguel, 2019*). The SIMPER profiles were generated based on the null hypothesis: (1) the taxon distribution was driven only by the number and breadth of available niche space, indicating the niche-based assembly; (2) the taxon distribution was dominated by the potential dispersal ability, regardless of the niche breadth under assemblage, indicating the dispersal-based assembly. Whether the empirical SIMPER results from niche- or dispersal-assembly processes can be evaluated by comparing the empirical SIMPER profile with the permutation SIMPER profiles (*Gibert & Escarguel, 2019*).

Recent studies suggested that CF in Maolan National Nature Reserve is far away from its native habitats, and the regional climax community of the natural reserve is subtropical karst evergreen and deciduous broad-leaved mixed forest (EDBM). Therefore, the occurrence of CF and its differences from EDBM have been an interesting issue for ecologists. Previous studies on *P. kwangtungensis* focused on population, community characteristics, interspecific interactions, limiting habitat factors, and its potential geographical distribution under global climate change (*Cui et al., 2012*; *Hou-Lin et al., 2007*; *Wang & Cui, 2023*; *Yuan-Zhi et al., 2006*). However, no study currently compared the biodiversity pattern and the underlying community assembly rules between CF and the regional climax community,

though these served as the crucial theoretical foundation for the situ conservation of *P. kwangtungensis*. Therefore, according to the standard manual (*Condit, 1998*) for the standard handbook of forest dynamic plot (FDPs) establishment, we established a series of FDPs to explore the abovementioned issues (*Wang & Cui, 2023*). By comparing the species composition and performing SAD fitting, we aimed to solve the questions: (1) to what extent did the CF differ from the EDBM in biodiversity pattern? (2) Whether the niche-based assembly processes determined the difference in species composition between CF and EDBM? (3) Whether the dispersal-based assembly processes determined the difference in species composition between CF and EDBM?

## MATERIALS & METHODS

### Study area and plot establishment

This study was conducted in Maolan National Nature Reserve, Yunnan-Guizhou Plateau, Southwest China. The reserve has a typical subtropical climate, with an annual temperature of 15.1 °C and annual rainfall of 1,374 mm. The highest temperature occurs in July (mean temperature of 15.1 °C), while the lowest is in January (5.2 °C). More than 80% of precipitation concentrates from May to October. Unlike typical subtropical evergreen broad-leaved forests, the karst area forms a different vegetation type with mixed evergreen and deciduous broad-leaved species due to its more heterogeneous habitats and poorer soil conditions. However, the evergreen coniferous forest featured by *P. kwangtungensis* usually distributes on the upper slopes due to the relatively less soil.

However, due to the greater than 50 m distance between each plot, which aims to avoid the confusing impacts of spatial autocorrelation, and *P. kwangtungensis* is normally distributed in habitats with higher altitudes or deeper slopes, we only established nine FDPs for both CF and EDBM.

To explore the difference in species diversity pattern between CF and EDBM and find out the driving factors of the difference, we received the field survey approval from the Maolan National Nature Reserve Administration of Guizhou Province and a series of FDPs were established in each vegetation type from September to November 2021. Since the least distance between any two sampling plots should be more than 50 m to avoid the impacts from spatial autocorrelation and ecotone, we established nine 20 m × 20 m FDPs in centralized distribution area of CF. In addition, to ensure the uniformity of repetition, we also established nine 20 m × 20 m FDPs in EDBM. Thus, a total of 18 20 m × 20 m FDPs were established in the study area (Table 1). In each FDP, the information of locations, elevation and soil conditions was recorded. All individuals in the plots were mapped, tagged, and identified to species, Furthermore, the diameter at breast height was measured to calculate species importance value (IV) (Eq. (1)), and determine vegetation community types.

$$IV = (R_f + R_d + R_{do})/3 \tag{1}$$

where $R_f$ is the relative frequency, $R_d$ is the relative density, $R_{do}$ is the relative significance.

**Table 1** Basic information of the established plots: *Pinus kwangtungensis* coniferous forest (CF); evergreen deciduous broad-leaved mixed forest (EDBM).

| Vegetation types | Number | Area (m²) | Elevation (m) | Slope | Latitude | Longitude |
|---|---|---|---|---|---|---|
| CF | 9 | 400 | 852–950 | 20°–60° | 25°35′35″–25°36′16″ | 107°42′21″–107°42′31″ |
| EDBM | 9 | 400 | 743–840 | 7°–32° | 25°11′46″–25°14′03″ | 107°54′55″–107°56′02″ |

To explore how the abiotic factors influence the species composition, we collected the top layer of soil (0–20 cm) without the litter at five points in each FDP, and the soil samples were mixed to determine the soil physicochemical properties. A total of five indices, including soil pH value, soil water content, soil total nitrogen content, soil total phosphorus content, and soil exchangeable calcium content were measured. The soil pH was obtained by measuring soil suspension through a pH meter (PH500T, INESA, Shanghai, China). The soil total nitrogen content and soil exchangeable calcium content were determined through Elemental Analyzers (UNICUBE trace, Elementar, Langenselbold, Germany). The soil total phosphorus content was determined spectrophotometrically at 700 nm by a continuous flow automated analyzer (AA3, Bran+Luebbe, Norderstedt, Germany).

### Differences in biodiversity patterns between CF and EDBM

Using the Margalef index, Shannon-Wiener index (Eq. (2)), Pielou's evenness index (Eq. (3)), and rarefied richness (sampled 10 individuals), a one-way analysis of variance (ANOVA) was performed to test the differences in α-diversity between CF and EDBM. Shannon-Wiener index ($H$) can indicate the diversity pattern of a community due to its sensitivity to variations in species richness and evenness of species abundance. Pielou's evenness index ($E$) is usually used to reflect the evenness of species abundance. Rarefied richness is used to characterize the species richness by controlling the confusing effects of sample size (*Heck Jr, Van Belle & Simberloff, 1975*). Additionally, the skewness of log-transformed species abundance depicts SAD symmetry and better quantifies the shape of SAD (*Magurran, 2005*). Usually, negative skewness refers to a higher proportion of rare species, while positive skewness reflects strong species dominance compared with a lognormal SAD (Fig. 1).

Metric multidimensional scaling (MDS) analysis was conducted on obtained SADs to examine the differences in species composition between communities (*Norden et al., 2009*). The similarity in species composition between types of FDPs was evaluated through the Chao-Jaccard abundance-based estimator (*Chao et al., 2005*). This approach can find a stable solution using several random starts, and standardizes the scaling in the result by a principal components rotation (*Oksanen et al., 2013*). This analysis was performed by using the "metaMDS" of the package "vegan" (*Oksanen et al., 2013*) in R3.4.2 (*R Development Core Team, 2022*).

$$H = -\sum_{i=1}^{S} P_i \ln P_i \qquad (2)$$

where $s$ is the total number of species in the plot; $P_i$ is the relative abundance of species $i$.

$$E = H/H_{\max} \qquad (3)$$

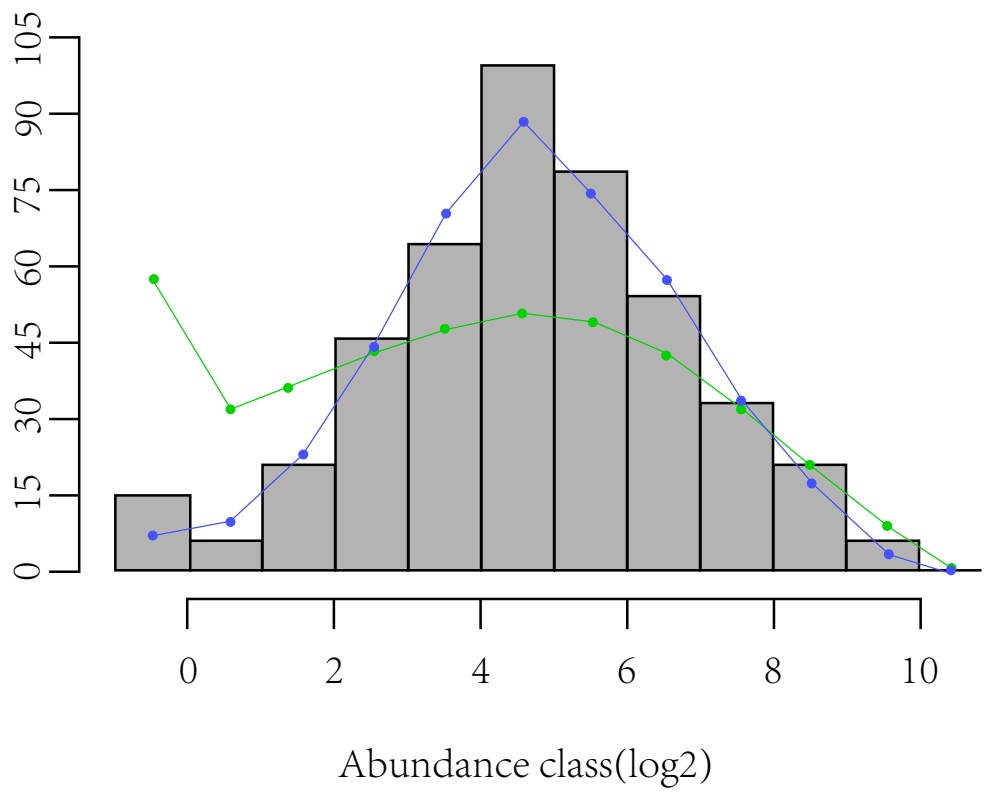

**Figure 1** **Display of skewness of log-transformed species abundance data depicts SAD symmetry.**

where $H$ is the actual Shannon-Wiener index while $H_{max}$ is the maximum of it, calculated by ln $s$.

## Examination of the first-order assembly process between CF and EDBM

Following the PER-SIMPER framework proposed by *Gibert & Escarguel (2019)*, this study examines the first-order assembly process influencing the dissimilarity in SADs between CF and EDBM. Based on *Clarke (1993)* SIMPER analysis, *Gibert & Escarguel (2019)* developed the PER-SIMPER analysis, which introduces a random permutation-based step. SIMPER analysis is a distance-based procedure that computes the relative contribution of each taxon to the OAD among groups of taxonomic assemblages. The Bray–Curtis coefficient for abundance was employed in SIMPER analysis to quantify the contribution of each taxon to the dissimilarity among groups of taxonomic assemblages (*Clarke, 1993*; *Legendre & Legendre, 1998*). Utilizing the empirical SIMPER pattern of a taxon locality occurrence data set, the PER-SIMPER analyzes 1,000 independently randomized occurrence data tables by fixing column or row to form the null SIMPER profile distribution (95% confidence intervals) of dispersal- and niche-assembly process. In addition, both sample richness and taxon uniquity remained unchanged under a maximally constrained null model. A third null SIMPER profile distribution was generated, corresponding to the null hypothesis
that both dispersal- and niche-assembly processes dominate the taxonomic assembly. The null distribution closest to (if not including) the empirical SIMPER profile indicates the primary driver contributing to the observed taxonomic occurrence structuring among groups (details about the null permutation can be found in the paper of *Gibert & Escarguel (2019)*). For each permutation model, the empirical SIMPER profile is compared with the null SIMPER profiles by computing the logarithm of the sum of squared deviations between two profiles (*E* is as follows):

$$E = \text{Log}_{10}\left(\sum_{i=1}^{i=p}(\overline{\gamma}_{i(\text{null})} - \overline{\gamma}_{i(\text{obs})})^2\right)$$

where $\overline{\gamma}_i = \frac{\delta_i}{\delta}$ is the contribution of the ratio of $\delta$ to the mean of $\delta$. Lower *E* indicates a closer similarity between the two compared SIMPER profiles. Thus, a PER-SIMPER analysis was conducted on SADs between CF and EDBM, using the "PER-SIMPER" function in R3.4.2 software (*R Development Core Team, 2022*).

### Determination of abiotic factors on α-diversity

Redundancy analysis (RDA) is a multivariate statistical approach based on principal component analysis that is used to examine the correlation between explanatory variables (*i.e.,* environmental factors) and response variables (*i.e.,* species composition). Abiotic factors could play a crucial various role in shaping community structure (*Xu et al., 2018*). Therefore, RDA was used to examine the influence of soil physicochemical properties on dissimilarities in species composition between CF and EDBM in this study. This analysis was performed with the "vegan" package in R3.4.2 (*R Development Core Team, 2022*).

## RESULTS

### Difference in α-diversity between CF and EDBM

CF had a total of 46 species and was dominated by *P. kwangtungensis*, with an IV of 0.43. EDBM had a total of 128 species and was dominated by *Boniodendron minus*, with an IV of 0.06 (File S2). By comparing species diversity between CF and EDBM, significant differences were found in the Shannon-Wiener index, Margalef richness, Pielou evenness, rarefied richness, and skewness (Fig. 2), indicating significant differences in both components (species richness and evenness) of α-diversity between CF and EDBM. Even after controlling the impacts of abundance, the species richness of EDBM was still significantly higher than that of CF (Fig. 2D). In addition, the EDBM showed higher species dominance compared to CF (Fig. 2E). MDS result also showed that the species composition of CF was significantly different from that of EDBM (Fig. 2F).

### First-order assembly processes driving dissimilarity in species composition between CF and EDBM

Results of PER-SIMPER analysis showed that the lowest E-values for dispersal- and niche-assembly profile, indicating that both niche-assembly and dispersal-assembly processes drove the dissimilarity in species composition between CF and EDBM (Fig. 3). Additionally,

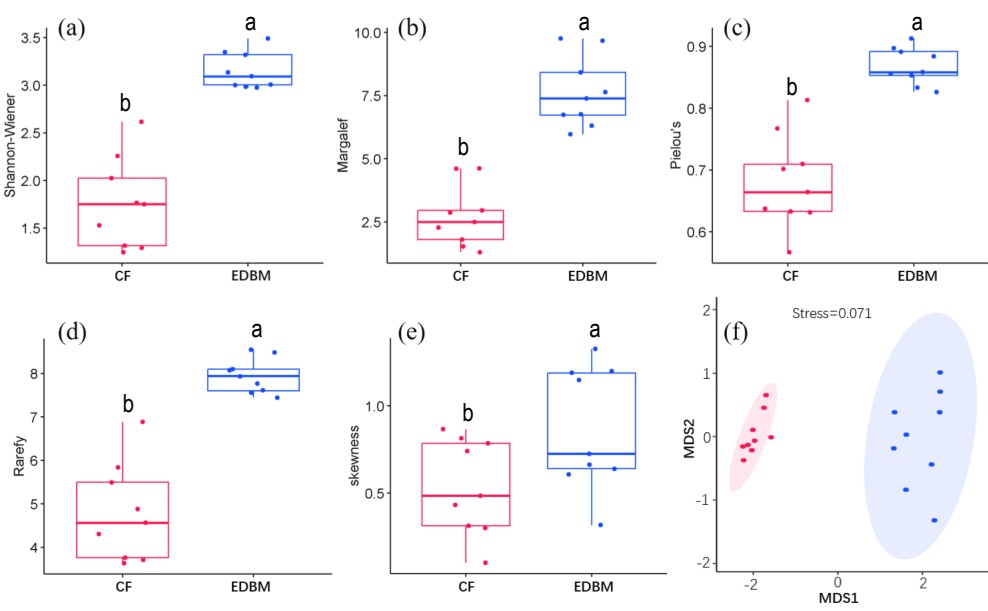

**Figure 2** α-**diversity between CF and EDBM.** (A) The Shannon–Weiner index; (B) the Margalef richness; (C) the Pielou Evenness; (D) the rarefied species richness; (E) the skewness between communities; (F) the species composition between communities. The red box reflected the *Pinus kwangtungensis*. coniferous forest (CF), while the blue box indicated the subtropical karst evergreen and deciduous broad-leaved mixed forest (EDBM). Significant differences were marked by different letters.

compared with niche-assembly process, the dispersal-assembly process had a lower E-values, indicating a more contribution of dispersal-assembly process to the dissimilarity in species composition.

## Abiotic factors influencing the α-diversity

Results of RDA showed that the soil physicochemical properties determined the difference in α-diversity between CF and EDBM (Fig. 4). Additionally, the soil exchangeable calcium content, soil total phosphorus content, soil water content, and soil pH value had influence on the difference in α-diversity. Specifically, soil exchangeable calcium content and soil total phosphorus content showed a positive relationship with Margalef index, and a negative relationship with the other α-diversity indices. Soil total nitrogen had a relatively minor impact on the community composition.

## DISCUSSION

This study compared the α-diversity, identified the first-order assembly process, and revealed the strong uniqueness in CF. However, both niche-assembly and dispersal-assembly process drove the dissimilarity in species composition between CF and EDBM, with the dispersal-assembly process making a greater contribution. The results indicate that there is a high degree of uncertainty in the occurrence of species within the CF community, particularly for rare species.

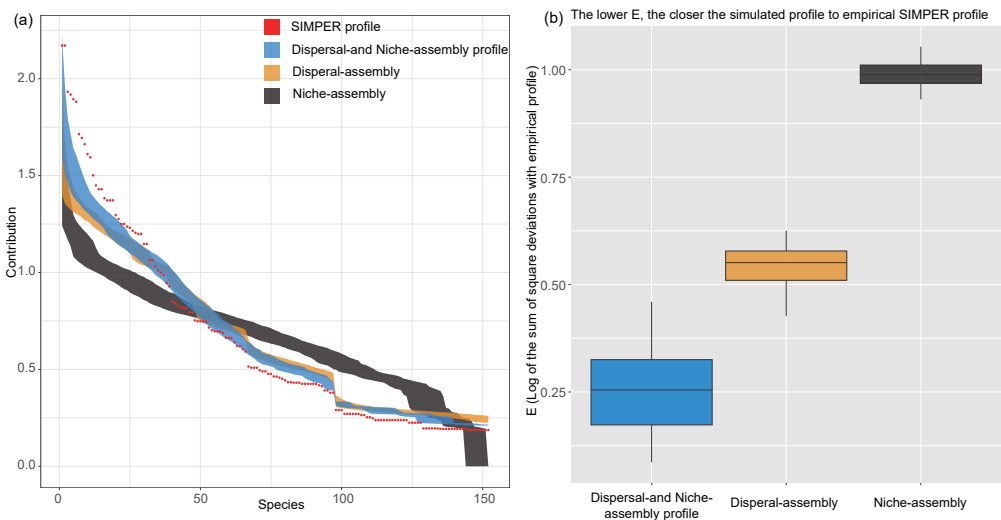

**Figure 3** **Results of PER-SIMPER analysis.** The orange box indicated the dispersal-assembly process, while the black box indicated the niche-assembly process. The blue box represented both processes dominating the first-order assembly process. Lower $E$-values showed a more significant influence of the process on the dissimilarities between groups of plots.

## Difference in α-diversity between CF and regional climax community

By comparing the CF with a typical regional climax community (EDBM), it was found that both the species richness and evenness of the CF were significantly lower (Fig. 2). The average individual density of the CF (0.297 individuals/m$^2$) was also lower than that of the EDBM (0.377 individuals/m$^2$). According to the species–energy hypothesis (*Akatov et al., 2023*), more individuals would increase species accumulation within the same regional scope, which might be one possible reason for the higher species richness of EDBM compared to CF (*Chu et al., 2019*; *McGill, 2011*). However, the rarefied richness of the EDBM was notably higher than that of the CF, indicating that the species richness of the CF remained lower even after controlling the cumulative effects of individuals. The biotic and abiotic filtering frameworks may play an important role (*Münkemüller et al., 2020*). The dominance of the CF was markedly lower than that of the EDBM, showing a high dominance of the EDBM (Fig. 3). The higher abundance and larger size of individuals compressed the ecological niche space of other species, indicating a much higher competitive ability of *P. kwangtungensis* compared to others, which limited colonization and survival of other species and formed monodominant species pattern (*Stanley Harpole & Tilman, 2006*; *Wang & Cui, 2023*). Similar patterns have been found in many evergreen coniferous forests, such as naturally occurring *Pinus roxburghii* forests in tropical regions, as well as artificially planted forests such as *Pinus massoniana*, *Cryptomeria japonica*, and *Cunninghamia lanceolata*, where evergreen coniferous species dominate and exhibit low species richness patterns (*Sloan, Zimmerman & Sabat, 2007*; *Wang & Cui, 2023*; *Zang et al., 2021b*). Meanwhile, the lack of subdominant species in CF allowed the colonization of rare species, contributing more to the difference in species composition

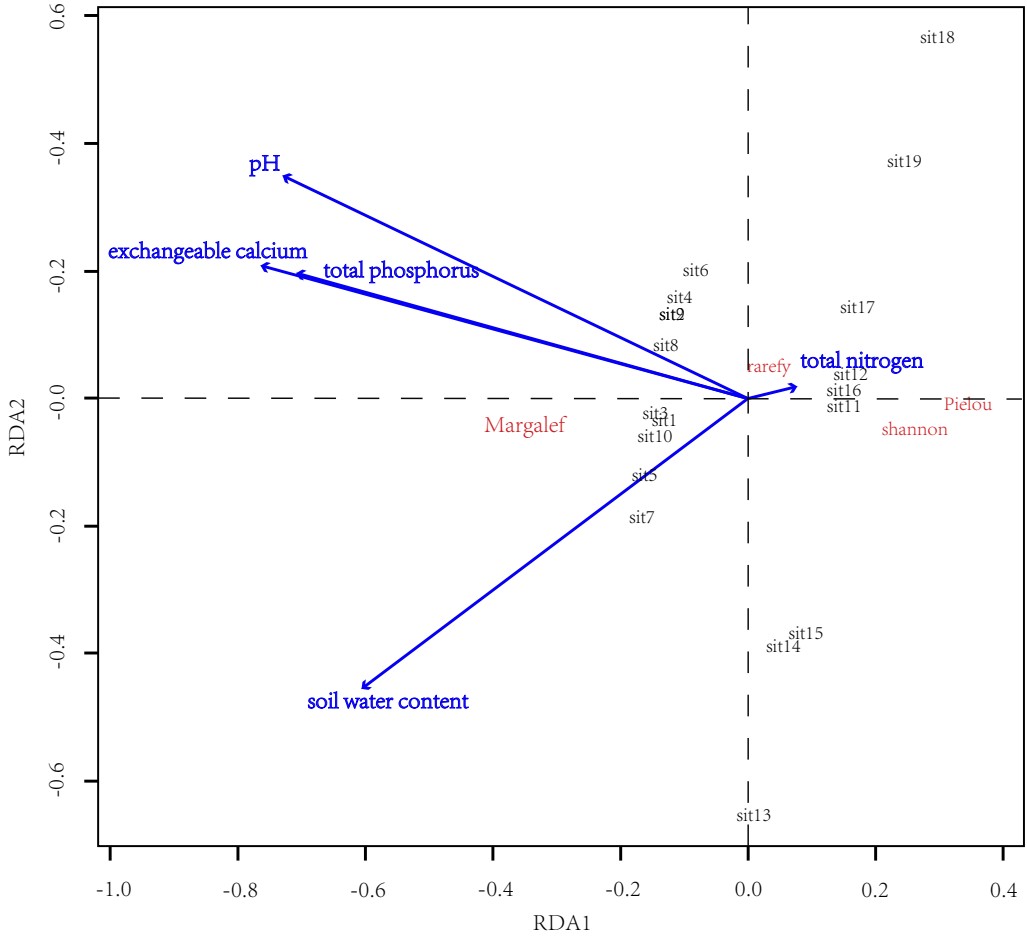

**Figure 4** **Results of RDA.** The α-diversity indices were denoted by red letters, while the groups of plots were shown in black letters. The blue lines indicated the soil physicochemical properties.

(*Lamanna et al., 2017*; *Zhang et al., 2014*). This phenomenon could also be a primary factor leading to the scarcity of species under such biotic conditions. Some researchers have pointed out that evergreen coniferous species commonly exhibit allelopathy, which could inhibit the establishment of other tree species, thereby reducing species richness (*Ehlers, Charpentier & Grøndahl, 2013*). Other studies also showed that poor light resources and soil conditions in evergreen coniferous forests could also be limiting factors for species diversity patterns. For example, low light resource quality in evergreen coniferous forests limits the establishment of shade-tolerant tree species (*Loke & Chisholm, 2023*; *Wang & Cui, 2023*). This study showed that the CF had almost no shade-tolerant pioneer tree species, which supported the view to some extent. In addition, the CF is shade-intolerant, similar to deciduous broad-leaved tree species in eco-strategy. Therefore, the presence of CF somewhat restricts the ecological niche space of deciduous broad-leaved tree species, reducing species richness (especially in broad-leaved tree species (*Cui et al., 2012*; *Wang &*

*Cui, 2023*)). In contrast, the EDBM is mixed with both evergreen and deciduous broad-leaved tree species. More deciduous broad-leaved tree species can increase species richness and evenness, resulting in a higher level of species diversity. Such diversity patterns also emphasized the importance of considering reducing deciduous tree species when performing *P. kwangtungensis* conservation to reduce the severe competition for lights. High habitat heterogeneity can also change species diversity patterns (*Bar-Massada, Kent & Carmel, 2014*; *Brown et al., 2013*). The habitat heterogeneity hypothesis suggests that increasing heterogeneity leads to an increase in small habitat types that can support plant growth and survival. The differing quality of small habitats can support the survival of different life forms, thereby maintaining a higher level of diversity (*Bar-Massada, Kent & Carmel, 2014*; *Franklin et al., 2013*). This study found that although both the CF and the EDBM were formed on limestone (as the parent rock), the microhabitat of the CF was relatively homogeneous without the presence of gullies, exposed rocks, or surface patterns. The EDBM exhibited higher levels of horizontal habitat heterogeneity, which provided a non-biological environmental foundation for maintaining diversity patterns and further promoted greater species diversity (*Bar-Massada, Kent & Carmel, 2014*; *Brown et al., 2013*; *Franklin et al., 2013*; *Wang & Cui, 2023*).

## First-order assembly process driving the dissimilarity in species composition

The results showed that both the CF and the EDBM were primarily driven by dispersal- and niche-assembly processes, with the stochastic process playing a slightly larger role in determining species composition differences (Fig. 3). This indicated that the community assembly of CF may be mainly controlled by stochastic processes relative to deterministic processes. The CF community exhibited monodominance with a sharp decline in the number of individuals of other species (Fig. 2E). Both results indicated that apart from *P. kwangtungensis*, the number of individuals of other tree species was extremely few, with a significant variation in presence between plots (Fig. 2F). This reflected the uncertainty of companion species in the CF, indicating strong stochasticity on the occurrence of other species (*Matthews et al., 2019*; *McGill, 2011*). From the perspective of species occurrence, neutral processes dominate the probability of occurrence of most species, especially the rare species. These results could also be driven by demographic, phylogenetic and biogeographic histories of individuals, populations and species. In contrast, the EDBM are mostly common species sharing the same regional species pool (*Chen et al., 2019*). Functional traits adapt to variations in abiotic factors, while convergence of a trait value suggests co-occurring species often appeared in similar abiotic conditions, leading to stronger habitat filtering (*Pappas, Fatichi & Burlando, 2016*; *Xu et al., 2018*). Therefore, changes in functional traits and abiotic factors may by one of the factors in community assembly process. Maolan National Nature Reserve is a typical karst area where strong habitat filtering effects and habitat heterogeneity restrict the dispersal and colonization of most species (*Gu et al., 2019*; *Wang et al., 2023*). When species disperse into this region, which species become companion species is largely randomized in CF. Thus, although the α-diversity of CF is

totally different from that of the EDBM, no certain evidence was found to support the necessity of such patterns.

### Abiotic factors driving the difference in α-diversity

The results of this study showed that the soil exchangeable calcium content and soil total phosphorus content were the main factors driving the difference in α-diversity between the CF and EDBM. Previous studies showed that the phosphorus content of soils was the limiting factor on species diversity in tropical or subtropical forests (*Wang et al., 2023*; *Zang et al., 2021a*). However, our results indicated that the soil total nitrogen content had no effect on the regional difference in species diversity among different vegetation types (Fig. 4). The varied speeds of biological cycles in different soil nutrients might be one possible reason (*Alvarez-Clare, Mack & Brooks, 2013*; *Wang et al., 2023*). The view of phosphorus limitation was mainly due to the relatively slow phosphorus cycle (*Alvarez-Clare, Mack & Brooks, 2013*). Previous studies showed that the soil available phosphorus came from rock decay, which was relatively slow and limited the speed of phosphorus cycling (*Laliberté et al., 2015*; *Zotz & Asshoff, 2010*). More phosphorus was fixed in the plant tissue with increasing biomass, decreasing the available phosphorus content (*Alvarez-Clare, Mack & Brooks, 2013*). Additional studies suggested that the $H_2PO_4^{2-}$ tended to form insoluble complexes with $Al^{3+}$ or $Fe^{3+}$ under acidic soils (*Laliberté et al., 2015*; *Zang et al., 2021a*). However, our previous studies on karst restoration showed no significant phosphorus limitation, as nitrogen cycling may be more limiting to nutrient content than phosphorus cycling (*Laliberté et al., 2015*; *Turner, Brenes-Arguedas & Condit, 2018*; *Wang et al., 2023*). Such nitrogen limitation also supported the idea that the EDBM still performed potential succession compared with subtropical evergreen broad-leaved forests. Thus, the soil nitrogen content might dominate the soil fertility, driving the species diversity patterns. In addition, higher calcium content has been one key component of soil physicochemical properties in karst areas. A growing body of studies showed that diversity had a higher dependence on the soil calcium content (*Batalha et al., 2015*; *Guo et al., 2019*; *Guo et al., 2017*; *Guo et al., 2015*), which is also clearly demonstrated in this study.

## CONCLUSIONS

*P. kwangtungensis* is an endangered species which has a narrow distribution. This study quantified the differences in species diversity and community assembly between the *P. kwangtungensis* coniferous forest and the evergreen and deciduous broadleaved forest within the Maolan National Nature Reserve, and found that the *P. kwangtungensis* coniferous forest had a lower species richness and evenness compared with the evergreen and deciduous broadleaved forest in the same reserve. However, such difference resulted possibly from both stochastic processes and deterministic processes with stochastic processes contributing more. Thus, our study suggested that the stochastic processes dominated the difference in species composition between the two vegetation types, indicating to a certain extent, the auxiliary species contributing less in shaping *P. kwangtungensis* community structure. Such patterns emphasized the importance of tree species selection in *P. kwangtungensis* conservation. In addition, our study suggested that

soil calcium and phosphorus contents played more important roles in driving species diversity pattern of *P. kwangtungensis* coniferous forest in local scales, emphasizing the importance of soil fertile in driving the species pattern and community assembly rules, as well as its growth, development and distribution. Therefore, we should consider more whether the soil nutrient conditions could well meet the requirements of *Pinus kwangtungensis* when conducting conservation projects.

Plant functional traits could be another effective perspective in determining community assembly rules, due to it is more sustainable to changes in environments. However, our study quantified the community assembly rules based on the species abundance distribution, which might lack of the information of how the traits respond to the various habitats. Further analysis should perform the verification both from the phylogenic and functional perspective.

## ACKNOWLEDGEMENTS

We thank the Maolan National Nature Reserve Administration of Guizhou Province for the assistance with the establishment of our forest dynamic plots, as well as ensuring our safety in the field. We thank Essentialslink Language Services for its linguistic assistance during the preparation of this manuscript. Lastly, we extend our appreciation to the reviewers for providing valuable feedback on the manuscript.

### Funding

This work was supported by the Science and Technology Foundation Program of Guizhou Provincial Health Commission (gzwkj2023-482) and the Guizhou Provincial Key Technology R&D Program ([2023]111). The funders had no role in study design, data collection and analysis, decision to publish, or preparation of the manuscript.

### Grant Disclosures

The following grant information was disclosed by the authors:
Science and Technology Foundation Program of Guizhou Provincial Health Commission: gzwkj2023-482.
Guizhou Provincial Key Technology R&D Program: [2023]111.

### Competing Interests

Yeheng Wang is employed by Zibo Real Estate Registration Center.

### Author Contributions

- Xingying Fan analyzed the data, prepared figures and/or tables, authored or reviewed drafts of the article, and approved the final draft.
- Longchenxi Meng analyzed the data, prepared figures and/or tables, authored or reviewed drafts of the article, and approved the final draft.
- Yeheng Wang analyzed the data, authored or reviewed drafts of the article, and approved the final draft.

- Lipeng Zang conceived and designed the experiments, analyzed the data, prepared figures and/or tables, authored or reviewed drafts of the article, and approved the final draft.

### Data Availability

Raw data showing species abundance in *Pinus kwangtungensis* and conifer-broadleaf forests are available in the Supplemental Files.

### Supplemental Information

Supplemental information for this article can be found online at http://dx.doi.org/10.7717/peerj.17899#supplemental-information.

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
