# Peer review of "Stochastic process drives the dissimilarity in biodiversity patterns between Pinus kwangtungensis coniferous forest and evergreen deciduous broad-leaved mixed forest in karst area"

_PeerJ, doi:10.7717/peerj.17899_

## Round 0.1 · original submission · Major Revisions

Although the reviewers recognized merits, they mention limitations and drawbacks, raising some misgivings about the way the manuscript has been written and analyzed. Overall (1) the introduction should be better structured, including a clear hypothesis, some predictions and the approaches used to test it, (2) data analyses need to be improved/clarified, and (3) a more in-depth discussion and a clear message about how the results can be used in conservation programs should be presented. Furthermore, ‘Data Availability’ statement is missing in the main text. Note that all data must be made available publicly in a permanent open repository and links, including a doi, should be provided in the Data Availability Statement. I hope that you will find all advice helpful when revising the manuscript.

**Language Note:** PeerJ staff have identified that the English language needs to be improved. When you prepare your next revision, please either (i) have a colleague who is proficient in English and familiar with the subject matter review your manuscript, or (ii) contact a professional editing service to review your manuscript. PeerJ can provide language editing services - you can contact us at [email protected] for pricing (be sure to provide your manuscript number and title). – PeerJ Staff

Reviewer 1 ·

Basic reporting

Xingying Fan et al., investigate patterns of biodiversity and the underlying community assembly rules between Pinus kwangtungensis forest community and conifer-broadleaf forest community. The methodology is adequated and in general, the results show clear differences between the two comunity evaluated (Figure 3), which is an interesting result. My main criticism to the manuscript is that currently is only descriptive. I think that in order to make it more appealing to readers of PeerJ, the authors should elaborate on the results obtained, and discuss the possible implications of their results from a biogeographic or even phylogenetic point of view.I propose some suggestions (see below).

Experimental design

In lines 138-140, what were the criteria for establishing the plots?
I am not sure if the number of forest dynamic plots (18) is sufficient to achieve consistent inferences. If it is true, the authors should justify that the number of plots is sufficient.
In lines 144-152 there are too many acronyms that are no longer used in the rest of the manuscript. I suggest leaving only those strictly necessary.

Validity of the findings

Results
In the section of results there is not reference to diversity data (Supplementary material), number of species, variation of species richness within each plot. What about of native species? What is the prevalence of species? What are the families more abundant?. I understand that there is a dominant species, but Why are not these results mentioned?
What was the goal to measure DBH and height? Is there any correlation between diameter and height?
Is there any relationship between the RDA analysis and the presence of determined species?
In the legend of figures, avoid the use of acronyms.

Discussion
In lines 228-231 the authors should mention in a pair of lines what are the implications of their findings.
In the lines 78-85 the authors mentioned that community structure may be result of demographic, phylogenetic and biogeographic histories. These factors are not clearly discussed in the section of discussion.
The authors should mention the possible role of the environment (Figure 5) forming the patterns of biodiversity evaluated.
Although it is not the goal of the study, the authors should discuss that these results could change when functional traits be evaluated.

Additional comments

I do not have any comments.

Reviewer 2 ·

Basic reporting

The paper deals with an extremely important topic for the preservation of endangered tree species, as in this case Pinus kwangtungensis. All praise to the authors for the efforts they made in the realization of this work.

Experimental design

Research question well defined, relevant and meaningful. It is stated how the research fills an identified knowledge gap. Forest dynamic plots (FDPs) are an excellent example of good practices for solving experimental questions in the natural sciences.

Validity of the findings

According to my insight into the submitted text, the results reached by the authors are very significant for the preservation of the researched species and the expansion and popularization of the species, its importance for the global scientific community.

Additional comments

I have no additional comments. I had difficulty with the writing style. But that's not a big complaint.

·

Basic reporting

General comment

1. This manuscript assesses the relevance of either the niche or the dispersal assembly hypotheses of taxonomic communities. If I correctly understand the hypothesis to be tested is if either assembly hypotheses explain each of the assemblages. It does not test if the two communities differ. There are two hypotheses and not just one if both communities are different. This aspect should be phrased in the whole manuscript.

Specific comments

Abstract

2. Lines 24-26, instead of describing the plots established it would be better if the hypothesis or main objective is stated here which is not described any where in the abstract. See comment 1.

3. Line 25, what is the meaning of Form.?

4. Line 33, it is necessary to find a better description for “determinacy” or a synonym

5. Lines 38-47, it is not clear how the study of conservation biology is relevant for the data presented in the manuscript. It appears reasonably that a high diversity community should be conserved when compared with a P. kwangtungensis dominated community. Another possibility is to test in each community the way in which they are structured, and which is presented in the rest of the introduction. In any case the manuscript should clearly distinguish among these two theoretical scenarios, community structuring and conservation biology and management, and explicitly connect them.

6. Lines 92-94, in this paragraph the introduction of the methodology used is relevant but should be presented with the hypotheses at hand, that is if the two communities are assembled following the niche or the dispersal models.

7. Line 113, substitute “suggested” with “suggest”

8. Lines 115-116, in different parts of the manuscript you mention that the two communities are “totally different” and is not clear to the reader if they do not have any species in common. If that is the case a quantitative assessment is needed

9. Lines 123-127, the two hypotheses should be restated. The first one pertains to comment 8. Are they “totally different” or how different are they which is the question posed. Why would they be similar. Is one (the one with P. kwangtungensis) a perturbed state from the other?. The second hypothesis relates to the difference in assembling processes in both communities but there are two different hypotheses, first if assemblage is controlled by niche and secondly if it is controlled by dispersal

Materials and Methods

10. Lines 138-143, it is not clear why nine plots were used. Did you test if this size would express any differences in the assembly processes?, also, did you use nine plots of EDBM and nine of CF?, please explain

11. Lines 205-207, please explain the basis of the RDA and how does the question asked relates to the assembly hypotheses

Results

12. In general there is a need to describe the results better

12. Line 211, sibtstitute “Weiner” with “Wiener”

13. Line 212 and 216, what do you mean by “complete difference” and totally different, see also comment 8

Discussion

14. Lines 228-231, apparently both the niche and the dispersal assembly processes partially explain community structure but the way in which the paragraph is presented suggests that it is only the dispersal assembly process which is contradictory

15. Line 236, refer to at least one publication related to the “species-energy hypothesis”

16. Lines 327-330, rephrase this part of the paragraph, see also comment 14

17. Line 330, what do you mean by similar stability, this concept has several interpretations and you have not mention it before

18. Figure 1, these maps are rather confusing, please mark the specific plots that were studied, you can also add a Table with the geographic data

19. Figure 2, it is not clear how this Figure fits in the whole manuscript, it is not explained in the results section nor discussed in the discussion section. Also a better description of the source of the data is needed. Is it from the nine plots?

20. Figure 3, you can delete “indicated”, “showed”, “reflected”, “displayed”, “showed” and “indicated

21. See comment 12

22. Figure 4, why is not there an orange box in part b)?

23. As commented in comment 11 explain the reason to include the RDA

Experimental design

See basic reporting

Validity of the findings

See basic reporting

---

## Round 0.2 · accepted · Accept

After careful consideration of the authors' revisions and their thorough responses to the reviewers' concerns, I am pleased to accept this manuscript in its current form.

Sincerely,
Alison

·

Basic reporting

The authors have made significant and clear changes that have improved the amnuscript.

Experimental design

No comment

Validity of the findings

No comment